# Nutritional Predictors of Mortality after 10 Years of Follow-Up in Patients with Chronic Kidney Disease at a Multidisciplinary Unit of Advanced Chronic Kidney Disease

**DOI:** 10.3390/nu14183848

**Published:** 2022-09-17

**Authors:** Guillermina Barril, Angel Nogueira, Graciela Alvarez-García, Almudena Núñez, Carmen Sánchez-González, Mar Ruperto

**Affiliations:** 1Advanced Chronic Kidney Disease Unit, Department of Nephrology, Hospital Universitario de la Princesa, C/Diego de León 62, 28006 Madrid, Spain; 2Department of Pharmaceutical & Health Sciences, School of Pharmacy, Universidad San Pablo-CEU, CEU Universities, Urbanización Monteprincipe, 28660 Madrid, Spain

**Keywords:** ACKD unit, advanced chronic kidney disease unit, bioelectric impedance analysis, body composition, chronic kidney disease, fluid overload, handgrip strength, nutritional follow-up, mortality, protein-energy wasting

## Abstract

Nutritional monitoring in advanced chronic kidney disease (ACKD) units provides personalized care and improves clinical outcomes. This study aimed to identify mortality risk factors in chronic kidney disease (CKD) patients on nutritional follow-up in the multidisciplinary ACKD unit. A retrospective cross-sectional observational study was conducted in 307 CKD patients’ stage 3b, 4–5 followed-up for 10 years. Clinical and nutritional monitoring was performed by malnutrition-inflammation score (MIS), biochemical parameters (s-albumin, s-prealbumin, and serum C-reactive protein (s-CRP), body composition measured by bioelectrical impedance analysis (BIA), anthropometry, and handgrip strength measurements. The sample was classified into non-survivors, survivors, and censored groups. Of the 307 CKD patients, the prevalence of protein-energy wasting (PEW) was 27.0% using MIS > 5 points, s-CRP > 1 mg/dL was 19.20%, and 27.18% died. Survivors had higher significant body cell mass (BCM%) and phase angle (PA). Survival analyses significantly showed that age > 72 years, MIS > 5 points, s-prealbumin ≤ 30 mg/dL, PA ≤ 4°, and gender-adjusted handgrip strength (HGS) were associated with an increased risk of mortality. By univariate and multivariate Cox regression, time on follow-up (HR:0.97), s-prealbumin (HR:0.94), and right handgrip strength (HR:0.96) were independent predictors of mortality risk at 10 years of follow-up in the ACKD unit. Nutritional monitoring in patients with stage 3b, 4–5 CKD helps to identify and treat nutritional risk early and improve adverse mortality prognosis.

## 1. Introduction

Chronic kidney disease (CKD) is estimated to rise from the 16th most common cause of death in 2016 to the 5th in 2040 [1], thus increasing the prevalence of CKD exponentially over the coming decades [2]. CKD has been recognized as an important risk factor for cardiovascular (CV) mortality and is a risk multiplier in a high percentage of patients with hypertension and diabetes mellitus (DM) [2].

Management of modifiable risk factors associated with fatal and non-fatal cardiovascular events (hypertension, dyslipidemia, poor glycemic control, obesity, and elevated proteinuria) showed a greater association with CKD progression and mortality outcomes, while their identification and correction effectively improved clinical outcomes [3,4].

One of the well-known CKD progression factors and independent predictors of mortality is protein-energy wasting syndrome (PEW), which can be higher than 50.0% in stage 5 CKD before starting renal replacement therapy (RRT) [5]. PEW is frequently associated with fluid overload at the expense of increased extracellular water (ECW), whether accompanied or not by excess total body water (TBW) when muscle mass (MM) has decreased in relation to the inflammatory state. The diagnostic criteria for PEW were described by the International Society of Renal Nutrition and Metabolism in 2008 [6], defining malnutrition in CKD as more than insufficient food intake but also associated with inflammation, both feeding back on each other and constituting PEW syndrome which, among other alterations, leads to a decrease in body compartments of fat and fat-free mass (FFM), including MM [5]. The diagnosis of PEW is based on several categories in which biochemical parameters such as serum albumin (s-albumin) and serum prealbumin (s-prealbumin), body mass index (BMI), MM, and protein intake are modified in the presence of an inflammation state.

As CKD progresses, accumulated uremic toxins, endocrine disturbances such as insulin resistance, metabolic acidosis, and inflammatory phenomena lead to anorexia and inadequate dietary intake, and increased energy expenditure, which is associated with changes in body composition [7]. Each of these uremic alterations, separately or together, leads to increased morbidity and mortality in CKD patients. To date, there is no single marker to define PEW, and therefore it is required to use several tools to assess nutritional status, combining laboratory parameters, body composition, and muscle strength measures, as well as validated and recommended nutritional screening tools such as the malnutrition-inflammation score (MIS) questionnaire [8,9,10].

Serum albumin (s-albumin) concentration as a universal nutritional marker has shown that levels below 3.8 g/dL were progressively associated with an increased mortality rate in CKD patients [11,12]. s-Prealbumin has been considered as a marker as a short-live marker in some published articles as well as a mortality predictor [13,14]. Furthermore, several studies have shown that s-CRP values in kidney patients > 1 mg/dL or >3 mg/dL were associated with lower survival [15,16,17].

Bioelectrical impedance analysis (BIA) is a validated and simple-to-perform method for measuring body composition and changes in hydration status in CKD and RRT. One of the main factors influencing outcomes is phase angle (PA), which has been established at a cut-off point of <4° in different research studies [18,19] as a predictor of mortality in CKD and dialysis patients.

Sarcopenia is a common condition characterized by the reduction in skeletal muscle strength and low amount muscle quantity or quality. Muscle strength is often more important in CKD patients than the amount of MM, because of which there may be a normal amount of muscle but with an altered fiber structure and low strength. Indeed, skeletal muscle strength has also been shown to be the first link to be considered in the diagnosis of sarcopenia according to the 2019 European consensus [20]. In various studies on CKD patients, the prevalence of sarcopenia ranges from 5.0% to 37.0% depending on the criteria used for diagnosis [21,22,23]. In older CKD patients, regular physical exercise prior to CKD or being physically active adapted to the characteristics of the CKD helps to prevent and/or improve multifactorial sarcopenia [24]. Therefore, it seems important to recognize nutritional risk factors in the development of CKD in order to emphasize those markers that could be predictive of mortality before starting RRT. The aim of this study was to identify nutritional predictors of mortality risk in non-dialysis patients with stage 3b-5 CKD followed for 10 years in a multidisciplinary advanced chronic kidney disease unit (ACKD unit).

## 2. Materials and Methods

### 2.1. Patient Population

A retrospective cross-sectional observational study was carried out in the multidisciplinary ACKD unit of the Hospital Universitario La Princesa (Madrid, Spain) from February 2012 to February 2022.

Patients older than 18 years were eligible if they had an estimated glomerular filtration rate (eGFR) ≤ 45 mL/min/1.73 m^2^ or CKD stage 3b-5 not on dialysis [1] and were in the ACKD unit for more than 3 months of follow-up before being included in the study. Exclusion criteria were CKD patients with acute processes (e.g., active infection), major surgery in the last 3 months, lower limb amputation, malignant tumor, nutritional support with enteral tube feeding or parenteral nutrition as well as life expectancy < 3 months due to any other cause.

The cause of death was categorized based on the International Classification of Diseases 10th edition (ICD-10) in the following areas: infectious diseases (including sepsis), cardiovascular diseases (CVD), systemic diseases (including diabetes, lupus, and rheumatoid arthritis), gastrointestinal diseases (including gastrointestinal bleeding), neurologic diseases (including stroke), musculoskeletal diseases, and other causes. Mortality data were retrospectively verified from medical records.

At baseline, 317 CKD patients were selected for this study, of which 10 patients did not meet the inclusion criteria. Patients were classified into three groups: survivors, no survivors, and censored CKD patients. The causes of censure among the participants were the onset of RRT (hemodialysis or peritoneal dialysis), renal transplantation, or transfer to another center for other reasons. A final sample size of 307 ACKD patients was included in this study (Figure 1).

This study was conducted by the Helsinki Declaration and Good Clinical Practice Guidelines. Ethical approval and permission to conduct this study were obtained from the local Ethics Committee (code no: 4257).

### 2.2. Data Collection

Socio-demographic parameters and clinical, nutritional, and laboratory data were retrospectively collected from each patient’s medical record. The main etiology of CKD and time in the ACKD unit (months) were recorded. The estimated glomerular filtration rate (e-GFR) was measured using the Chronic Kidney Disease Epidemiology Collaboration (CKD-EPI) equation [25]. In the ACKD Unit, a comprehensive and multifactorial approach to the patient’s risk factors for progression is taken at each medical visit, including modifiable risk factors for CKD progression (control of proteinuria, uric acid), as well as CV risk factors (DM, hypertension, dyslipidemia), control of anemia and bone mineral metabolism, and underlying comorbidities. In addition, as an integral component of the care and treatment of CKD patients, nutritional status is usually assessed every 3 months in conjunction with scheduled medical care. A diet with a protein intake of 0.8 g/kg body weight/day and dietary control of salt, phosphorus, and potassium according to the dietary guidelines for patients with CKD is usually recommended through personalized nutritional counseling in the ACKD unit [11]. Adherence to dietary recommendations (a diet low in protein, saturated fat, and salt, as well as low in potassium and phosphorus and phosphorous/protein ratio) is routinely monitored in all patients at each medical visit by e-GFR, proteinuria, and glycosylated hemoglobin in diabetic patients, as well as blood pressure measurements and assessment of serum sodium and potassium levels. Dyslipidaemia is also assessed by lipoprotein profile and markers such as uric acid as potentially modifiable CV risk factors in adherence to dietary treatment in CKD patients. Likewise, CV risk factors are treated pharmacologically by individualized medical prescription of antihypertensive drugs, diuretics, lipid-lowering agents, antiplatelet agents, or oral anticoagulants. In diabetic patients, treatment with oral antidiabetic agents and/or insulin is prescribed accordingly to e-GFR, proteinuria, and serum potassium levels as components of the comprehensive approach and multifactorial intervention in the ACKD unit.

Normalized protein nitrogen appearance (nPna) as a surrogate indicator of dietary protein intake according to the equation proposed by the Kidney Disease Outcomes Quality Initiative clinical practice guidelines was used [10].

### 2.3. Malnutrition-Inflammation Score

The MIS questionnaire [8,9] was used within the routine nutritional care protocol to assess nutritional status in the outpatient medical ACKD unit. This screening tool includes ten components: seven subjective (obtained from medical history and physical exams), and three objective parameters (s-albumin, total binding iron capacity, and body mass index (BMI). The MIS components range from 0 (normal) to 30 (severely wasted). Although the well-nourished patients were considered from 0 to 2, in agreement with other studies, PEW was defined as an MIS ≥ 5 points [26,27].

### 2.4. Anthropometric Measures and Body Composition Analysis

Body weight (BW) and BMI were recorded as anthropometric measures. The handgrip strength (HGS) of the right hand was measured with a hand-held dynamometer (Baseline^®^ mod.12-0240). The measurements were taken in triplicate and the mean of the values obtained was recorded.

Body composition was evaluated with a bioelectrical impedance analyzer (BIA-101^®^, Akern-RJL Systems, Florence, Italy) at a frequency of 50 kHz and 800 μA constant electrical flow through the body in the horizontal supine position. Tetrapolar distal validated method [28] by using two pairs of disposable electrodes (BiatrodesTM 100’S, Akern, Florence, Italy) placed on the right hand and foot and whole-body BIA was performed. In patients with an arteriovenous fistula in the dominant arm, anthropometric and muscle strength measurements were carried out on the contralateral side. BIA-derived measurements such as PA, exchangeable Na/K, TBW, ECW, intracellular water (ICW), body cell mass (BCM), FFM, and MM were analysed by Bodygram Pro V.3.0 software based on manufacturer’s standards. In accordance with previous studies in CKD patients [27,29,30], the cut-off point to be wasted for PA was <4°. According to the sarcopenia consensus [20] for classifying HGS measures normal cut-off points for women < 16 kg/m^2^ and men < 27 kg/m^2^ were used in this study. Various studies [31,32,33] demonstrated that low values of HGS were significant predictors of survival in CKD.

### 2.5. Laboratory Parameters

Laboratory samples were retrospectively collected from medical records. S-albumin was analyzed by the colorimetric standard method (Roche/Hitachi 904^®^/Modular ACN 413) using the bromocresol green method [34]. Based on the diagnostic criteria for PEW [6], the cut-off point for s-albumin was on at 3.8 g/dL. s-Prealbumin and s-CRP were measured by immunoturbidimetry methods (Roche/Hitachi 904^®^/Model P: ACN 218, Roche Diagnostics, Basel, Switzerland). The concentration of s-CRP (*non-high sensitivity*) was considered as an inflammatory marker set on a cut-off point at or above 1 mg/dL.

### 2.6. Statistical Analysis

Continuous variables were expressed as mean and standard deviation, with the comparisons between groups performed by the ANOVA-factor test for normally distributed variables. Categorical variables were defined using proportions and were analyzed by the Chi-squared test. Correlations were tested by bivariate Pearson correlations for continuous variables. The Kaplan–Meier method [33] was used to calculate cumulative survival probabilities according to the different cut-off points (age: 72 years; MIS: 5 points; s-prealbumin: 30 mg/dL; PA: 4°, and HGS gender-adjusted). The difference between survival curves was assessed by the *log-rank test*. Univariate and multivariate Cox proportional hazard regression analyses were used to evaluate the prognostic value of the different parameters and the risk of death. The variables that significantly affected survival in the univariate analysis were subsequently tested in multivariate models using a forward stepwise procedure with a probability to the entry of 0.05 and a probability of removal of 0.20. The multivariate Cox proportional hazard model was adjusted for age and gender as potential confounders. The hazard ratio (HR) and 95% confidence interval (95%CI) were calculated. Statistical analyses were performed by using SPSS version 23.0 (IBM Corp., Armonk, NY, USA) software. *p*-values < 0.05 were set as significant.

## 3. Results

### 3.1. Global Data and Comparison between Groups

Of the 307 patients with CKD who participated in this study, 61.90% were men, the mean age was 70.16 ± 12.46 years. The global prevalence of DM was 46.30% in the sample, being the primary etiology leading to CKD in 23.0% of the patients in this study. Sixty-two CKD patients (20.19%) died at 10 years of follow-up. Eighty-four patients with CKD (27.30%) were censored as the main cause of onset of RRT. The causes of death were CVD (*n* = 21; 34.40%), cerebrovascular disease (*n* = 5; 8.19%), infection (*n* = 9; 14.75%), neoplasm (*n* = 9; 14.70%), and unknown cause (*n* = 13; 21.0%). The demographic and clinical characteristics of the 307 ACKD patients are summarized in Table 1.

According to CKD stage, 30 patients were in CKD stage 3b (9.80%), while there were 164 patients in CKD stage 4 (53.40%) and 113 patients in CKD stage 5 without dialysis (36.80%). The mean time in the ACKD unit was 58.89 ± 37.59 months. The mean value of e-GFR was significantly higher in survivors than in no-survivors and censored CKD patients (*p* = 0.001), while nPna did not differ between the three groups. The total MIS values were 4.14 ± 3.19 points, being higher in non-survivors (MIS: 4.54 ± 3.08 points) than in survivors (MIS: 3.88 ± 3.12 points) or censored patients (MIS: 4.34 ± 3.39 points) (*p* = 0.30). By analyzing MIS ≥ 5 points as PEW criteria, was significantly much higher in non-survivors than in survivor or censored groups (Table 1). MIS was positively correlated with age (*r* = 0.268; *p* = 0.01), s-CRP (*r* = 0.185; *p* = 0.01), TBW (*r* = 0.252; *p* = 0.01) and negatively with s-prealbumin *(r* = −0.229; *p* = 0.01), ICW (*r* = −0.226; *p* = 0.01), BCM% (*r* = −0.186; *p* = 0.001), PA (*r* = −0.238; *p* = 0.01) and right-HGS (*r* = −0.132; *p* = 0.05).

### 3.2. Anthropometric Measures, Body Composition Analysis, and Laboratory Data

Table 2 shows body composition and laboratory data of no survivor, survivor, and censored groups. BW and BMI did not differ significantly between groups. The mean value BMI was consistent with overweight in all three groups analyzed. According to BMI classification, 38.10% were overweight while 26.10% were obese. Overweight was found more commonly in males (41.30%) vs. females (30.50%), whereas obesity was similar in both genders (male: 25.40%; female: 27.30%). Survivors were overweight (37.80%) and obese (29.80%) compared to non-survivors (40.30% and 20.90%, *respectively*).

Survivors had significantly lower exchangeable Na/K and ECW than non-survivors (*at least, p* < 0.05), while higher significant measures of ICW%, BCM%, and PA were also found in the survivor group. Body composition measurements such as FM, FFM, and MM tended in a non-significant manner to be higher in survivors. Overall, it should be noted that 47.40% had dynapenia. Mean right-HGS values were significantly lower in non-survivors than in the survivor or censored groups. Analyzing by gender, in male non-survivors, 57.10% had a right-HGS < 27 kg/m^2^, while in survivors, it was 27.40% (*p* = 0.046). In females, right-HGS was < 16 kg/m^2^ in 80.80% of non-survivors and in 66.70% of survivors (*p* = 0.001).

Globally, 32.30% of patients had MIS ≥ 5 points, mean s-albumin values of 3.94 ± 0.58 g/dL, and s-CRP values of 1.13 ± 1.90 mg/dL. The mean s-albumin concentration was 4.10 ± 0.43 g/dL, with no significant differences between the groups. S-albumin concentration < 3.8 g/dL was found in 17.12% of survivors and 19.35% of non-survivors (*p* = 0.99). s-Albumin was positively correlated with ICW% (*r* = 0.372; *p* < 0.001), BCM% (*r* = 0.304; *p* < 0.001), right-HGS (*r* = 0.159 *p* = 0.005), and nPna (*r =* 0.281; *p* < 0.001) and negatively with age (*r* = −0.213; *p* < 0.001), TBW% (*r* = −0.257; *p* < 0.001), ECW% (*r* = −0.372; *p* < 0.001), exchange Na/K (*r* = −0.431; *p* < 0.001), MIS (*r* = −0.467; *p* < 0.001), and s-CRP (*r* = −0.315; *p* < 0.001).

Mean s-prealbumin levels were slightly lower in non-survivors compared to survivors (25.83 ± 5.21 vs. 28.61 ± 6.60) or censored (29.53 ± 7.81 mg/dL) groups (*p* = 0.005) along with low levels of s-CRP between groups (*p* > 0.05). s-Prealbumin was positively correlated with age (*r* = 0.380; *p* < 0.001), ICW% (*r* = 0.392; *p* < 0.001), BCM% (*r* = 0.284; *p* < 0.001), and right-HGS (*r* = 0.248; *p* < 0.001) and negatively with MIS (*r* = −0.225; *p* < 0.001) and s-CRP *(r =* −0.303; *p* < 0.001). Only 16.1% of non-survivors, 19.30% of survivors, and 21.40% of censored patients had s-CRP levels > 1 mg/dL (*p* = 0.724).

### 3.3. Factors Predicting Mortality

The results of the survival curve analyses showed mean age (>72 years.), MIS > 5 points, PA ≤ 4°, s-prealbumin ≤ 30 mg/dL, and the lowest gender-adjusted HGS cut-off point using right-HGS were predictors of mortality in patients with stage 3b, 4–5 CKD, as shown in Figure 2 and Figure 3.

### 3.4. Cox Proportional Hazards Analysis of Mortality

The univariate Cox proportional hazards analysis results showed that parameters such as age, time on ACKD unit, MIS, or body composition measurements including exchangeable Na/K, ECW%, ICW%, BCM%, PA, right-HGS as well as s-albumin and s-prealbumin were predictors of mortality risk at 10 years of follow-up (Table 3).

Using Cox multivariate proportional hazards analysis, after adjusting for potential confounders (age and gender), several parameters, such as follow-up time in the ACKD unit, s-prealbumin, and right-HGS were significantly associated with the risk of all-cause mortality in CKD patients (Table 4).

## 4. Discussion

The results of this study demonstrate that regular clinical and nutritional care and follow-up using a combination of nutritional, body composition, and functional parameters contribute to improved survival of CKD patients during 10 years of follow-up in a multidisciplinary ACKD unit. In this study, patients with ACKD were older, more often male, and were mostly in CKD stages 4–5 (90.50%). The main etiology leading to CKD was DM (23.0%), including an overall high proportion of diabetics diagnosed in the study for other causes (46.30%). Follow-up time in the ACKD unit was longer than one year in 75.20% of patients. In fact, in the specific monographic ACKD Unit, patients with CKD were assessed and followed up medically and nutritionally from the perspective of comprehensive management of modifiable risk factors for CKD progression and cardiovascular risk (DM, hypertension, dyslipidemia, proteinuria, and uric acid), as well as control of anemia, bone mineral metabolism, underlying comorbidities, and nutritional status, as components of the comprehensive and multifactorial approach to the CKD patient. Notably, a multicentre, randomized, open-label, multicentre clinical trial in diabetic patients with CKD [35] compared the intensive multifactorial treatment of mortality risk factors (blood pressure control, glycated hemoglobin, total cholesterol, LDL and HDL) with standard medical care on major fatal/non-fatal CV events. At 13 years of follow-up, the study demonstrated that only the strategy based on the intensive treatment of major risk factors was effective in reducing all causes of mortality by 47.0% [35]. Results from this study showed that CVD, cerebrovascular disease and DM among other risk factors were the main causes of death in CKD patients, in accordance with previous studies. The main causes of death in diabetics were cardiovascular events (4.20%), cancer (4.20%), and infections including sepsis (3.50%) in this study.

In the current study, the evolution of CKD patients was analyzed considering non-death causes for leaving the study: onset of RRT, transfer or loss to follow-up, or living donor renal transplantation. In these scenarios, follow-up until the event was considered, taking into account the main features of the censored, surviving, and non-surviving groups in the 307 CKD patients analyzed.

In the present study, it was found that 20.19% of CKD patients had died with a mean follow-up in the ACKD unit of 32.82 ± 24.93 months. The leading causes of death were CVD, cancer, and infections. In fact, age ≥ 72 years or over was a significant risk factor for mortality (Figure 2). Older age is inherently a contributory factor to adverse outcomes in CKD associated with increased comorbidities (e.g., CVD) and a higher risk of PEW and inflammation with declining kidney function [36]. It should be noted that PEW can be triggered in any age range and is closely related to the acute intercurrent processes and multiple comorbidities that often accompany patients with CKD.

PEW is a common condition in patients with CKD and a recognized predictor of mortality associated with CVD. The results of this study showed that PEW, as measured by MIS score ≥ 5 points, was 27.0% in CKD patients followed for 10 years. These findings are remarkably lower than those found in other similar studies [37] in CKD stages 3b, 4–5. A low prevalence of PEW 28.0% is observed in the early stages of CKD, which gradually increases as CKD progresses and even reaches 70.0% in CKD stage 5 [5,38].

The MIS questionnaire is a validated and useful nutritional screening tool that has been shown to be effective as an independent predictor of mortality in CKD and dialysis patients [39]. In this study, non-survivors had a significantly higher frequency of MIS ≥ 5 points compared with survivors or censored groups. MIS was positively associated with age, TBW, and s-CRP and inversely with BCM (%), PA, s-prealbumin, and right-HGS (at least, *p* < 0.05). In addition, MIS ≥ 5 points were as shown to be a risk factor in the Kaplan–Meier analysis (Figure 2, Table 3) and a significant predictor of mortality in the univariate analysis (HR: 4.38; CI95%: 1.005–1.161; *p* = 0.036). In the ACKD unit, the periodical monitoring of nutritional status provides early identification and treatment of PEW and avoids its evolution to severe and irreversible grades associated with inflammation such as cachexia [5,9,26,27,40]. Conversely, mean BMI values were associated with overweight in this study. By analyzing BMI categories, non-survivors were more obese 29.80% compared with survivors who only 21.0% were obese. Previous studies [41,42] have shown in stage 3–5 CKD that overweight and mild obesity as measured by BMI provides advantages in terms of CKD progression and overall survival, whereas a BMI value < 23 kg/m^2^ increased the risk of mortality. Moreover, mean BMI values were associated with overweight, although without significant differences with cumulative survival in this study.

BIA is a simple and non-invasive technique for measuring body composition and hydration status in CKD and dialysis patients. In the current study, BIA-derived measurements (BCM, PA), as well as hydration indicators (exchangeable Na/K, ECW, ICW), were significantly different between survivor and non-survivor groups (*all, at least, p* < 0.05) (Table 2). In CKD patients, low BCM was related with early PEW, reduction in MM, and decreased HGS [43]. In addition, PA is a cell-integrity predictor of morbidity and mortality in CKD patients. Significant mean PA value differences were found between non-survivors and survivors. It is worth noting that the median PA in this study was 4°. A cut-off point of PA ≥ 4° has been shown to be a protective indicator of nutritional status in CKD and dialysis patients [44,45,46]. The results of this study showed that PA ≥ 4° was found in 41.90% of non-survivors and 53.40% of survivors, which was significantly demonstrated by Cox regression survival curves and univariate analysis. Clinical research studies on CKD [44,47,48] showed that PA was positively associated with s-albumin and s-prealbumin concentrations, FFM from BIA, and HGS along with CV events or mortality. The findings of this study are also consistent with those of previous studies in which PA ≥ 4° was shown to be a predictor of nutritional risk and mortality in CKD and dialysis patients [18,19,47]. In addition, Kaplan–Meier survival analysis revealed that patients with PA > 4° had significantly longer survival, as shown in Figure 3.

The HGS is an indirect measure of skeletal muscle performance, as well as being one of the commonly recognized diagnostic criteria for sarcopenia. Results from this study showed that significant differences with right-HGS adjusted by conventional gender cut-offs were found between non-survivors compared with the survivor and censored groups. Male and female non-survivors had significantly lower HGS values than survivors, with 32.90% of non-survivors having normal values, as well as 60.90% of survivors and 51.20% of censored survivors. In fact, the predictive ability of the HGS cut-off points (male: 27 kg/m^2^ or female: 16 kg/m^2^) was significantly confirmed in the survival curves (*long-rank test = 8.243*; *p <* 0.004) (Figure 3c). Furthermore, the results obtained in univariate and multivariate analyses showed that HGS was a significant protective factor for mortality at 10 years of follow-up. A recent study of 8767 patients with CKD [31] reported that 9.70% were likely to be sarcopenic, while the 10-years survival probability was 85.0% in sarcopenic patients compared to non-sarcopenic patients with CKD (89.0%).

S-albumin is an independent predictor of morbidity and mortality in ACKD and dialysis patients [11,12,14,17]. In this study, mean s-albumin levels did not differ significantly between no survivors and survivor groups. s-Albumin was ≥3.8 g/dL in 80.6% of non-survivors, 80.7% of survivors, and 81% of censored CKD patients. Univariate analysis showed that s-albumin was an independent risk factor for mortality, but no significant association was found in the multivariate analysis. As a possible explanation for these findings, it should be noted that the mean s-albumin levels in this study were 4 g/dL, being these results consistent with a low frequency of PEW and inflammation in the study. In addition, s-prealbumin was tested as an early marker of protein and inflammatory profile. S-prealbumin was significantly different between groups, being found at prealbumin levels > 30 mg/dL in 25.80% of non-survivors and 50.3% of the survivors (39.30% censored). Kaplan–Meier analysis showed that a s-prealbumin > 30 mg/dL was a significant survival factor, a finding also supported by multivariate Cox regression at a 10-year follow-up (Figure 3b). The findings of this study are in agreement with previously published studies [11,12,13,14,17] in which both s-albumin and s-prealbumin were found to be markers of morbidity and mortality in CKD and dialysis patients. 

Some strengths and weaknesses of this study should be noted. To our knowledge, this is the first study in patients with ACKD that evaluates various predictors of nutritional risk with mortality in a sample of patients with stage 3b, 4–5 CKD with follow-up for 10 years. So far, studies in ACKD stages 4–5 analyzing the predictive ability of various nutritional indicators of mortality risk are still scarce. However, this study is limited by the fact that it was performed in a single ACKD unit, and therefore, possible biases may have interfered with the study results. Only one measurement of each of the nutritional parameters and indicators was taken into account retrospectively. The data were extracted retrospectively from the nutritional history of each patient considering a minimum time > 3 months in the ACKD unit and taking into account that the patient had at least three consecutive medical and nutritional visits for data collection. In addition, drug therapy (antihypertensives, diuretics, lipid-lowering agents, antiplatelet agents, oral antidiabetic agents, and/or insulin) and some markers of CV risk, anemia control, and bone metabolism, as well as other associated comorbidities, were not included among the laboratory parameters collected in this study. The monitoring of these clinical factors was a part of the routine comprehensive and multifactorial management of patients in the ACKD unit. Thus, a combination of nutritional, inflammatory, and body composition parameters was used in accordance with the aim of the study. Nutritional assessment and periodic follow-up together with nutritional counseling and evaluation of adherence to dietary recommendations (low content of protein, saturated fat, sugars, phosphorous, phosphorous/protein ratio, potassium, and free-salt diet) are part of the comprehensive care and attention of the CKD patient in the ACKD unit. As a limitation, in this study, the assessment of dietary intake was not included, although indirect surrogate markers such as nPna, blood pressure control as well as serum sodium and potassium levels were used to assess adherence to individualized nutritional recommendations in CKD patients. Nevertheless, a combination of nutritional, inflammatory, and body composition parameters were widely used, although drug treatment or dietary intake was not recorded in the study. Furthermore, in clinical practice, achieving normal albumin values (≥3.8 g/dL) due to adequate dietary intake and/or low inflammation may be limiting for the identification of certain classical risk factors such as muscle wasting or sarcopenia. Thus, regular monitoring of nutritional status was carried out by a multidisciplinary team composed of a nephrologist, a nurse, and a nutritionist as part of the comprehensive and personalized care in the ACKD unit.

## 5. Conclusions

In conclusion, this study suggests that nutritional monitoring in patients with stage 3b, 4–5 CKD helps to identify and treat nutritional risk early and improve adverse mortality prognosis. Factors such as age and CKD itself favor PEW while time on CKD and clinical and nutritional monitoring, together with s-prealbumin levels and HGS measures, were all potentially modifiable parameters that reduced mortality in CKD stages 3b-5 during 10 years of follow-up in the multidisciplinary ACKD unit. Further studies are needed to define the impact of nutritional monitoring on mortality risk in ACKD patients on non-dialysis.

## Figures and Tables

**Figure 1 nutrients-14-03848-f001:**
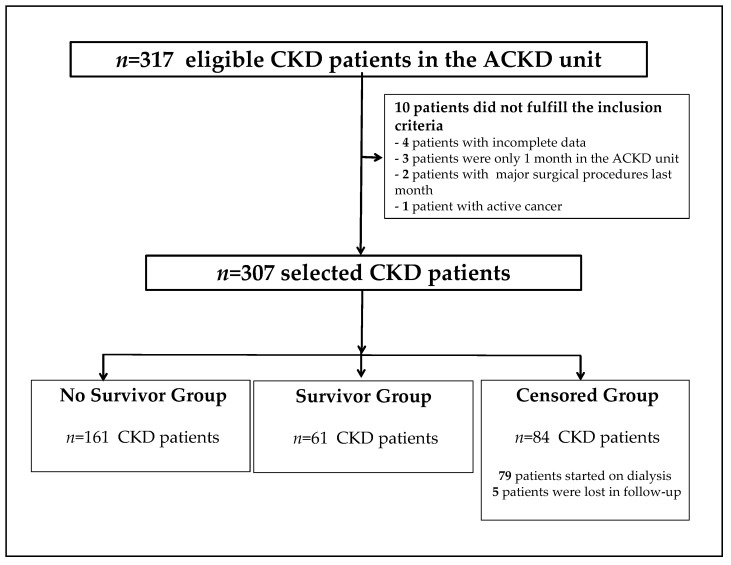
Flowchart of the recruitment and selection of chronic kidney disease patients in the study. ACKD unit, advanced chronic kidney disease unit; CKD, chronic kidney disease.

**Figure 2 nutrients-14-03848-f002:**
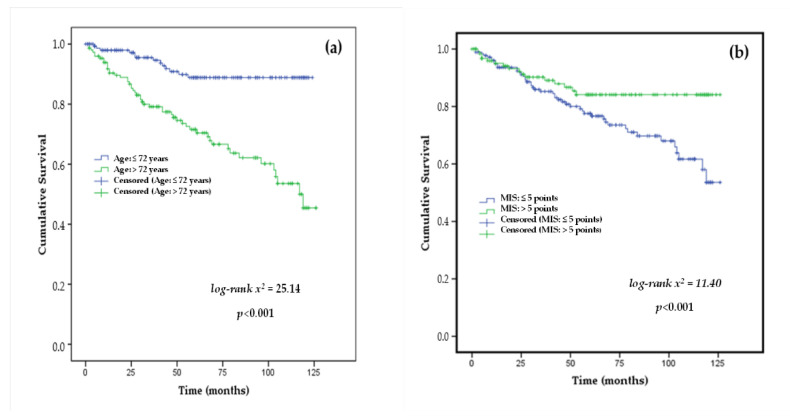
Kaplan–Meier survival curves of the study participants stratified by the cut-off-points of (**a**) age (≤72 vs. >72 years) and (**b**) malnutrition-inflammation score (MIS: ≤5 vs. >5 points) as mortality risk predictors in stage 3b, 4–5 chronic kidney disease patients.

**Figure 3 nutrients-14-03848-f003:**
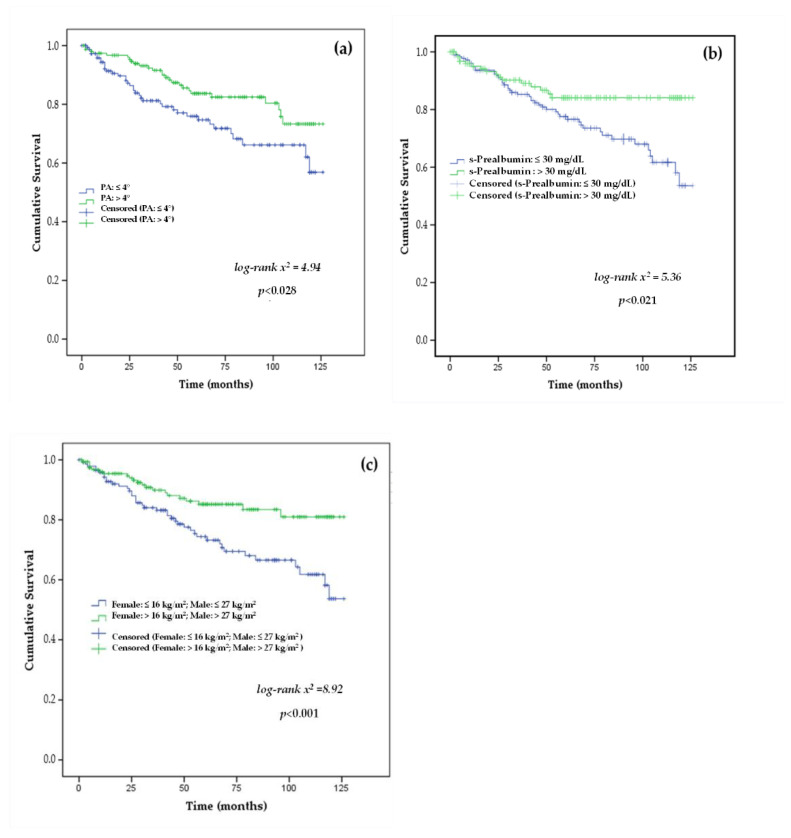
Kaplan–Meier survival curves of the study participants stratified by the cut-off-points of (**a**) phase angle (PA: ≤4 vs. >4°), (**b**) s-Prealbumin (≤30 vs. >30 mg/dL), and (**c**) handgrip strength in males (HGS: ≤26 vs. >26 kg/m^2^) and in females (HGS: ≤17 vs. >17 kg/m^2^) as mortality risk predictors in stage 3b, 4–5 chronic kidney disease patients.

**Table 1 nutrients-14-03848-t001:** Demographic and clinical characteristics of 307 advanced chronic kidney disease patients.

Variables	Global *n* = 307	No Survivors *n* = 62	Survivors *n* = 161	Censored *n* = 84	*p*-Value
Male *n*; (%)	212 (61.90)	36 (58.10)	113 (70.20)	63 (75.0)	0.003
Age (years)	70.16 ± 12.46	68.72 ± 12.43	77.55 ± 9.66	67.10 ± 12.33	<0.001
DM *n* (%)	142 (46.30)	28 (45.20)	76 (47.20)	38 (45.20)	0.940
Time ACKD unit (mo.)	58.89 ± 37.59	32.82 ± 24.93	64.09 ± 63.29	23.63 ± 19.93	<0.001
e-GFR (mL/min/1.73 m^2^)	18.96 ± 8.24	19.38 ± 7.75	20.78 ± 8.85	15.16 ± 5.82	<0.001
nPna (g/kg/day)	0.92 ± 0.24	0.90 ± 0.20	0.94 ± 0.28	0.90 ± 0.20	0.608
PEW * *n*(%)	83 (27.0%)	24 (38.70%)	37 (23.0%)	22 (26.20)	0.049

*p*-Values are based on Chi-square or ANOVA factor tests. * PEW, protein-energy wasting was assessed by malnutrition-inflammation score (MIS) questionnaire equal to or higher than 5 points. ACKD-unit, advanced chronic kidney disease unit; DM, diabetes mellitus; e-GFR, estimated glomerular filtration rate; nPna, normalized protein nitrogen appearance.

**Table 2 nutrients-14-03848-t002:** Anthropometric, body composition, and laboratory parameters in 307 advanced chronic kidney disease patients.

Variables	Global *n* = 307	No Survivors *n* = 62	Survivors *n* = 161	Censored *n* = 84	*p*-Value
BW (kg)	75.03 ± 16.35	71.61 ± 15.18	76.04 ± 17.17	75.60 ± 15.38	0.181
BMI (kg/m^2^)	27.04 ± 5.12	27.14 ± 5.20	27.67 ± 5.24	27.09 ± 4.88	0.634
Exchangeable Na/K	1.44 ± 0.49	1.57 ± 0.58	1.37 ± 0.45	1.47 ± 0.49	0.019
TBW (%)	53.59 ± 6.93	53.82 ± 8.08	52.91 ± 6.34	54.70 ± 7.04	0.152
ECW (%)	58.66 ± 8.27	56.66 ± 7.29	56.16 ± 8.57	56.61 ± 8.54	0.018
ICW (%)	42.99 ± 8.41	40.33 ± 7.29	43.81 ± 8.58	43.38 ± 8.54	0.019
BCM (%)	38.74 ± 9.37	36.51 ± 8.25	40.19 ± 9.79	37.62 ± 8.95	0.014
PA (°)	4.10 ±1.16	3.72 ± 0.97	4.24 ± 1.19	4.13 ± 1.19	0.012
FM (%)	32.11 ± 10.36	31.94 ± 9.91	32.58 ± 9.15	29.97 ± 9.57	0.123
FFM (%)	68.35 ± 9.48	67.88 ± 10.36	67.67 ± 8.99	70.01 ± 9.58	0.168
MM (%)	33.18 ± 7.88	32.02 ± 7.39	33.30 ± 8.21	33.81 ± 7.57	0.386
Right-HGS (kg/m^2^)	25.78 ± 9.90	21.04 ± 7.72	27.19 ± 10.27	26.51 ± 9.64	<0.001
s-Albumin (g/dL)	4.10 ± 4.47	4.10 ± 0.39	4.21 ± 0.44	4.15 ± 0.45	0.231
s-Prealbumin (mg/dL)	28.30 ± 6.83	25.83 ± 5.21	28.61 ± 6.60	29.53 ± 7.81	0.005
s-CRP (mg/dL)	0.82 ± 1.49	0.59 ± 0.73	0.98 ± 1.86	0.71 ± 1.01	0.158

*p*-Values are based on ANOVA-factor test. BCM, body cell mass; BMI, body mass index; BW, body weight; ECW, extracellular water; FFM, fat-free mass; FM, fat mass; ICW, intracellular water; MM, muscle mass; PA, phase angle; Right-HGS, right handgrip strength; s-Albumin, serum albumin; s-CRP, serum C-reactive protein; s-Prealbumin, serum prealbumin; TBW, total body water.

**Table 3 nutrients-14-03848-t003:** Univariate Cox proportional hazards analysis of all-cause mortality risk in stage 3b, 4–5 chronic kidney disease patients.

Predictor Variable	HR (95%CI)	*p*-Value
Gender	1.530 (0.926 to 2.528)	0.097
Age (years)	1.071 (1.071 to 1.101)	<0.001
DM *n*; (%)	0.161 (0.549 to 1.486)	0.688
Time ACKD-unit (months)	0.978 (0.969 to 0.986)	<0.001
e-GFR (mL/min/1.73 m^2^)	1.008 (0.979 to 1.038)	0.587
nPna (g/kg/day)	0.529 (0.156 to 1.794)	0.307
MIS (points)	1.080 (1.005 to 1.161)	0.036
BW (kg)	0.989 (0.973 to 1.005)	0.178
BMI (kg/m^2^)	0.997 (0.949 to 1.047)	0.899
Exchangeable Na/K	1.669 (1.169 to 2.385)	0.005
TBW (%)	1.004 (0.968 to 1.041)	0.838
ECW (%)	1.056 (1.026 to 1.087)	<0.001
ICW (%)	0.947 (0.920 to 0.975)	<0.001
BCM (%)	0.968 (0.942 to 0.995)	0.021
PA (°)	0.650 (0.513 to 0.824)	<0.001
FM (%)	0.397 (0.982 to 1.035)	0.529
FFM (%)	0.377 (0.966 to 1.018)	0.539
MM (%)	0.969 (0.937 to 1.002)	0.068
Right-HGS (kg/m^2^)	0.946 (0.919 to 0.973)	<0.001
s-Albumin (g/dL)	0.592 (0.355 to 0.987)	0.045
s-Prealbumin (mg/dL)	0.936 (0.900 to 0.973)	0.001
s-CRP (mg/dL)	0.954 (0.762 to 1.194)	0.680

BCM, body cell mass; BMI, body mass index; BW, body weight; DM, diabetes mellitus; ECW, extracellular water; e-GFR, estimated glomerular filtration rate; FFM, fat-free mass; FM, fat mass; ICW, intracellular water; MIS, malnutrition-inflammation score; MM, muscle mass; nPna, normalized protein nitrogen appearance; PA, phase angle; Right-HGS, right handgrip strength; s-Albumin, serum albumin; s-CRP, serum C-reactive protein; s-Prealbumin, serum prealbumin; TBW, total body water.

**Table 4 nutrients-14-03848-t004:** Multivariate Cox proportional hazards analysis of all-cause mortality risk in stage 3b, 4–5 chronic kidney disease patients.

Predictor Variable	HR (95% CI)	*p*-Value
Time of follow-up (months)	0.975 (0.966 to 0.984)	<0.001
s-Prealbumin (mg/dL)	0.946 (0.911 to 0.982)	0.003
Right-HGS (kg/m^2^)	0.953 (0.925 to 0.983)	0.002

*p*-Value was adjusted for age and gender as confounding factors. HR, hazard ratio; Right-HGS, right handgrip strength.

## Data Availability

Not applicable.

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
