# Peer review of "Nutritional Predictors of Mortality after 10 Years of Follow-Up in Patients with Chronic Kidney Disease at a Multidisciplinary Unit of Advanced Chronic Kidney Disease"

_nutrients, 2022, doi:10.3390/nu14183848_

Round 1
Reviewer 1 Report
This single-center observational retrospective study showed, during a 10-year follow-up, that nutritional monitoring in patients with stages 3b, 4 and 5 CKD can help identify and treat nutritional risk early, improving adverse mortality prognosis.
The paper is very interesting and quite original. However, the limitations of the study were only partially highlighted by the authors.
Therefore, this reviewer raises some issues that need to be addressed by the authors.
1- In the limitations of the study, line 421, the authors write "... drug treatment or dietary intake was not recorded in the study". In fact, in this study the risk was assessed only through nutritional predictors. This is the main issue that needs to be properly addressed in both the discussion and the section on limits. In patients with advanced CKD it is well documented that both composite cardiovascular outcome (MACEs) and mortality are strictly conditioned by multifactorial drug therapy. This statement is even more valid in the diabetic population, which represents 45% of this study, as was recently observed in the NID-2 study, a multicenter clinical trial of multifactorial intervention (Cardiovasc Diabetol (2021) 20:145. doi: 10.1186/s12933-021-01343-1), in which, among other things, the intake of salt in the diet was one of the risk factors to be treated in the intensified treatment arm. Therefore, I underline that this problem should be adequately addressed by the authors.
2- Actually, management of risk factors in subjects with advanced CKD is a very hot issue. However, in real life these patients, despite the worst renal and cardiovascular prognosis, are at high risk of being under-treated independently of the type of clinical setting. (J Hypertens. 2006 Aug;24(8):1655-61. doi: 10.1097/01.hjh.0000239303.93872.31.). This issue and above reference should be addressed in discussion.
3- A linguistic revision by a native English speaker is required.
Author Response
Answers to reviewer #1's comments and suggestions on the article entitled "Nutritional predictors of mortality after 10 years of follow-up in patients with chronic kidney disease at a multidisciplinary unit of advanced chronic kidney disease” by the authors G. Barril et al. (Nutrients code number 1885804).
Comments and Suggestions for Authors
- This single-center observational retrospective study showed, during a 10-year follow-up, that nutritional monitoring in patients with stages 3b, 4 and 5 CKD can help identify and treat nutritional risk early, improving adverse mortality prognosis.
The paper is very interesting and quite original. However, the limitations of the study were only partially highlighted by the authors.
Answer. Thank you in advance for your valuable comments and suggestions to improve our manuscript. The changes are highlighted in yellow in the second version of the manuscript.
-Therefore, this reviewer raises some issues that need to be addressed by the authors.
- In the limitations of the study, line 421, the authors write "... drug treatment or dietary intake was not recorded in the study". In fact, in this study, the risk was assessed only through nutritional predictors. This is the main issue that needs to be properly addressed in both the discussion and the section on limits. In patients with advanced CKD, it is well documented that both composite cardiovascular outcome (MACEs) and mortality are strictly conditioned by multifactorial drug therapy. This statement is even more valid in the diabetic population, which represents 45% of this study, as was recently observed in the NID-2 study, a multicenter clinical trial of multifactorial intervention (Cardiovasc Diabetol (2021) 20:145. doi: 10.1186/s12933-021-01343-1), in which, among other things, the intake of salt in the diet was one of the risk factors to be treated in the intensified treatment arm. Therefore, I underline that this problem should be adequately addressed by the authors.
Answer. Thanks for your comments and suggestions. In relation to cardiovascular (CV) risk factors, although this study analyzes nutritional parameters, the ACKD unit provides a comprehensive and multifactorial approach to the factors classically considered to be CV risk factors. Blood pressure is usually monitored at each medical visit and treated with antihypertensive drugs (ACE inhibitors, ARA2, calcium antagonists, and β-blockers) according to the characteristics of the patient and considered in case of overhydration the association with diuretics (furosemide, torasemide and/or spironolactone if CV risk with congestive heart failure in combination with furosemide). In addition, nutritional assessment and nutritional counseling are performed jointly in the medical visit by a nutritionist with expertise in renal disease.
As mentioned in the section on material and methods, in our ACKD unit we routinely provide nutritional counseling to control the intake of protein, salt, phosphorus, and potassium in the diet. A salt-free diet is recommended for all patients as an essential component of the approach to CV risk factors, especially in diabetics. Only in patients with salt-losing nephropathy, supplemental salt intake is recommended if necessary. In diabetic patients with CKD, nutritional counseling is personalized according to glycemic control, CV risk factors, and the stage of CKD.
Likewise, monitoring of CV risk factors includes control of dyslipidemia and whether treatment with lipid-lowering drugs (statins, ezetimibe or evolocumab if familial hypercholesterolemia or non-responders to statin therapy) is required, together with proteinuria. Additionally, given the age of the patients, in our advanced age unit and the presence of multiple comorbidities, antiplatelet agents (acetylsalicylic acid, clopidogrel) as well as other oral anticoagulants (warfarin, rivaroxaban, apixaban) are prescribed. Additionally, as we have commented in the study results, CVD was the main cause of mortality (34.40%). In diabetic patients on treatment with oral antidiabetic agents and/or insulin, treatment is intensified by establishing a therapeutic objective level of glycated hemoglobin ≤ 7% taking into account the type of DM, stage of CKD, proteinuria, and serum potassium levels. Therefore, although the perspective of the current study analyzes nutritional risk factors in patients with CKD, multifactorial pharmacological treatment of CV risk factors is routinely performed in the ACKD unit. In the second version of the manuscript, this important aspect is mentioned in lines 41 to 45 of the introduction section as well as the bibliographic references (3, 4 and 35) and in the discussion (Lines 358 to 364), and limitations (459 to 475) sections. Thank you again for your valuable comments.
2- Actually, management of risk factors in subjects with advanced CKD is a very hot issue. However, in real life these patients, despite the worst renal and cardiovascular prognosis, are at high risk of being under-treated independently of the type of clinical setting. (J Hypertens. 2006 Aug;24(8):1655-61. doi: 10.1097/01.hjh.0000239303.93872.31.). This issue and above reference should be addressed in discussion.
Answer. Thanks for your comment. Our manuscript highlights the importance of closely monitoring nutritional status, since there are mortality predictors related to it, although there is no doubt that this area, sometimes not sufficiently considered, should also be part of an integrated treatment of CKD in which the proper control of risk factors should also be monitored.
As mentioned in the previous point, in the ACKD monographic unit, the patient is periodically monitored during the medical visit that includes risk factors for progression of CKD (uric acid, obesity, proteinuria) and also CV risk factors (DM, hypertension, dyslipidemia), control of anemia and bone mineral metabolism, as well as the underlying comorbidities.
Given the high prevalence of patients with DM either as a cause of CKD or as comorbidity, control of glycemia and CV factors associated with DM can prevent the progression of CKD and its associated side effects. In addition, periodic monitoring of nutritional status is performed, which is often not sufficiently considered or treated in nephrology departments despite its importance in the progression of the disease and as an associated risk factor for mortality. Therefore, it is important to emphasize the importance of the care and monitoring of risk factors in CKD as well as the role of the multidisciplinary monographic ACKD units in the clinical setting from the perspective of the comprehensive approach to CKD and its treatment to guarantee optimal attention and high-quality care in the prevention, delaying the progression of CKD and the adequacy of treatment that provides quality of life in the renal patient.
3- A linguistic revision by a native English speaker is required.
Answer. The second version of the manuscript has been revised by a native English speaker. Thanks for revision.
Reviewer 2 Report
Barril G, Nogueira A., Alvarez G., Nuñez A. Sanchez C. and Ruperto M retrospectively investigated predictors of mortality risk in patients with advanced CKD in Madrid Spain for 10 years. The aim is very important and interesting, but less novelty. There are some questions and comments as below.
1. It considered to be the most important that timing of data collection in this study. I could not find the timing when Socio-demographic parameters, clinical, nutritional, and laboratory data were collected from each patient's medical record.
2. In the Figure 2,3, there are four different line columns in the figures, but only two are actually listed. In addition, the Figure3 does not match the explanation in the legends. It may need to be corrected.
Minor comment
The authors should use abbreviations appropriately. For instance, PEW in the abstract section does not spell out. "malnutrition-inflammation score (MIS) , estimated glomerular filtration (eGFR), C-reactive protein (CRP) and so on " are mentioned repeatedly, across several sections. Carefully review the entire document for inappropriate use of abbreviations.
Author Response
Answers to reviewer #2 comments and suggestions on the article entitled "Nutritional predictors of mortality after 10 years of follow-up in patients with chronic kidney disease at a multidisciplinary unit of advanced chronic kidney disease” by the authors G. Barril et al. (Nutrients code number 1885804).
Barril G, Nogueira A., Alvarez G., Nuñez A. Sanchez C. and Ruperto M retrospectively investigated predictors of mortality risk in patients with advanced CKD in Madrid Spain for 10 years. The aim is very important and interesting, but less novelty. There are some questions and comments as below.
Answer. Many thanks for your comments and suggestions to improve our manuscript. The changes are highlighted in yellow in the second version of the manuscript.
- It considered to be the most important that timing of data collection in this study. I could not find the timing when Socio-demographic parameters, clinical, nutritional, and laboratory data were collected from each patient's medical record.
Answer. Thanks for your comment. Data are collected from each patient's visits in the ACKD unit. They are usually performed at least quarterly and the interval is variable, adapting more frequently according to the patient's needs. Sociodemographic parameters, clinical, nutritional, and laboratory data were collected retrospectively from the medical records of each patient, taking into account that the inclusion criterion for the study was that the patients were in the CKD unit for 3 months or more. Likewise, for the selection of the data, it was assessed that the patient had at least 3 complete and consecutive medical and nutritional visits since the beginning in the CKD unit. Thank you for your question, this important point is included in the second version of the manuscript in the material and method (Lines 135 to 156) and in the discussion (Lines 459 to 462) sections.
- In the Figure 2,3, there are four different line columns in the figures, but only two are actually listed. In addition, the Figure3 does not match the explanation in the legends. It may need to be corrected.
Answer. Thanks for your comment. Due to an error in the manuscript, three of the figures did not appear in the file uploaded to the Nutrients platform. This error has been corrected in the second version of the manuscript.
Minor comment
The authors should use abbreviations appropriately. For instance, PEW in the abstract section does not spell out. "malnutrition-inflammation score (MIS), estimated glomerular filtration (eGFR), C-reactive protein (CRP) and so on " are mentioned repeatedly, across several sections. Carefully review the entire document for inappropriate use of abbreviations.
Answer. Thank you very much for helping us to improve our manuscript. According to your comment, the entire manuscript has been thoroughly revised and modified in the second version of the manuscript.
Round 2
Reviewer 1 Report
No further comments.
Reviewer 2 Report
The manuscript has been revised well. I think this manuscript will be acceptable